# A genome-wide association study (GWAS) of the personality constructs in CPAI-2 in Taiwanese Hakka populations

**Pei-Ying Kao[1], Ming-Hui Chen[2], Wei-An Chang[3,4], Mei-Lin Pan[3], Wei-Der Shu[3], Yuh-Jyh Jong[5,6,7], Hsien-Da Huang[5,8], Cheng-Yan Wang[9], Hong-Yan Chu[4], Cheng-Tsung Pan[8], Yih-Lan Liu[9]\*, Yeong-Shin Lin[1,10]\***

1 Institute of Bioinformatics and Systems Biology, National Yang Ming Chiao Tung University, Hsinchu, Taiwan, 2 Department of Hakka Language and Social Science, National Central University, Taoyuan, Taiwan, 3 Department of Humanities and Social Sciences, National Yang Ming Chiao Tung University, Hsinchu, Taiwan, 4 Research Center for Humanities and Social Sciences, National Yang Ming Chiao Tung University, Hsinchu, Taiwan, 5 Department of Biological Science and Technology, College of Biological Science and Technology, National Chiao Tung University, Hsinchu, Taiwan, 6 Graduate Institute of Clinical Medicine, College of Medicine, Kaohsiung Medical University (KMU), Kaohsiung, Taiwan, 7 Departments of Pediatrics and Laboratory Medicine, KMU Hospital, Kaohsiung, Taiwan, 8 Institute of Bioinformatics and Systems Biology, National Chiao Tung University, Hsinchu, Taiwan, 9 Institute of Education, National Yang Ming Chiao Tung University, Hsinchu, Taiwan, 10 Department of Biological Science and Technology, College of Biological Science and Technology, National Yang Ming Chiao Tung University, Hsinchu, Taiwan

\* elaineliu@nycu.edu.tw (YLL); yslin@nycu.edu.tw (YSL)

## Abstract

Here in this study we adopted genome-wide association studies (GWAS) to investigate the genetic components of the personality constructs in the Chinese Personality Assessment Inventory 2 (CPAI-2) in Taiwanese Hakka populations, who are likely the descendants of a recent admixture between a group of Chinese immigrants with high emigration intention and a group of the Taiwanese aboriginal population generally without it. A total of 279 qualified participants were examined and genotyped by an Illumina array with 547,644 SNPs to perform the GWAS. Although our sample size is small and that unavoidably limits our statistical power (Type 2 error but not Type 1 error), we still found three genomic regions showing strong association with Enterprise, Diversity, and Logical vs. Affective Orientation, respectively. Multiple genes around the identified regions were reported to be nervous system related, which suggests that genetic variants underlying the certain personalities should indeed exist in the nearby areas. It is likely that the recent immigration and admixture history of the Taiwanese Hakka people created strong linkage disequilibrium between the emigration intention-related genetic variants and their neighboring genetic markers, so that we could identify them despite with only limited statistical power.

## Introduction

The genome-wide association studies (GWAS) have been commonly used for identifying disease-related single nucleotide polymorphisms (SNPs) in human populations (e.g., Manolio [1]; Pearson & Manolio [2]). Comparing with the bi-parental quantitative trait loci (QTL)

**Data Availability Statement:** Data cannot be shared publicly because of the ethical regulation issued by the Research Ethics Committee for Human Subject Protection of National Chiao Tung University/IRB (reference number: NCTU-REC-109-

029EF). Data are available from the Research Ethics Committee for Human Subject Protection of National Chiao Tung University/IRB (contact via mail:rec@nctu.edu.tw) for researchers who meet the criteria for access to confidential data.

**Funding:** Chang WA was the project director, and this project was financially supported by the Higher Education Sprout Project of the National Chiao Tung University and Ministry of Education (MOE), Taiwan [Grant numbers: 104W977, 107W311, 108W21, and 109W29]. The members of the project include: Kao PY, Chen MH, Chang WA (director), Pan ML, Shu WD, Jong YJ, Huang HD, Chu HY, Liu YL, and Lin YS. The URL of the funder website is: https://sprout.moe.edu.tw/SproutWeb/Home/Index/en The funder had no role in study design, data collection and analysis, decision to publish, or preparation of the manuscript.

**Competing interests:** The authors have declared that no competing interests exist.

linkage mapping method, the linkage disequilibrium (LD) mapping utilizing GWAS provides more insights into the molecular genetic basis [3]. Recently, the application has also been extended to the studies of psychology and personality (e.g., Montag et al. [4]). With these advanced techniques, geneticists could study not only the association between a certain personality and a certain gene as decades ago (e.g., Ebstein et al. [5]; Schinka et al. [6]), but the associations among various personality traits and various SNPs in the human genome (e.g., de Moor et al. [7]; Kim et al. [8]; Koshimizu et al. [9]; Luciano et al. [10]). Moreover, van den Berg et al. [11] within the Genetics of Personality Consortium applied Item-Response Theory (IRT) to harmonize Neuroticism and Extraversion measures from different inventories. The method and the enlarged datasets were thus utilized to perform GWAS analyses by some following studies (e.g., de Moor et al. [12]; Lo et al. [13]; van den Berg et al. [14]). Not limited to the Big Five personality traits (Openness, Conscientiousness, Extraversion, Agreeableness, and Neuroticism; also known as the Five-Factor Model, FFM), Kim et al. [15] further studied six Neuroticism distinct facets to reveal their pathway-based associations.

The questionnaires utilized by the above GWAS studies involve some traditional ones like the Eysenck Personality Questionnaire [16], and mostly the FFM-related NEO Personality Inventory like Neo PI3 [17] or NEO-PI-R [18]. Although some of the aforementioned GWAS studies have included Eastern (oriental) samples to perform the analyses (e.g., Kim et al. [8]; Kim et al. [15]), the adopted personality questionnaires were actually developed based on Western theories of personality with Western samples. Originally when these measures were developed, it was assumed that personality traits are stable dispositions that are biologically based, and therefore they are applicable across cultures. However, personality traits are interdependent or mutually constitutive of environment or cultures [19]. Although dispositional traits have biological bases, and cultures don't change the individuals' genetic make-up, cultures may influence the ways dispositional traits are elaborated or reinforced during development and articulated or manifested across settings. Therefore, the assessment of personality across cultures would incorporate the measurement of universal and culture-specific traits [20].

To deal with this problem, Cheung et al. [21] regarded that the combination of emic-etic methods is a powerful way to integrate universal and culture-specific aspects of constructs or theories with a balanced treatment. They believed that cross-cultural comparison would be more significant if the instruments are designed to capture similar phenomena in different cultures as well as the characteristics culturally relevant to the local settings. The Chinese Personality Assessment Inventory (CPAI) [21] was developed by adopting the etic and emic approach in the Chinese context. CPAI consists of both etic personality construct similar to the constructs covered by the main Western personality theory, and emic personality constructs that are essential in the Chinese context. Given that CPAI is a lack of Openness construct, CPAI-2 was then revised based on CPAI by incorporating the Chinese indigenous openness concept [22]. A systematic cross-cultural comparison of the factor structures from Chinese samples (e.g., Hong Kong, Mainland China, and Taiwan), from other Asian samples (e.g., Japan, South Korea, Singapore, and Vietnam), and from Western samples (e.g., American, Dutch, and Romanian) consistently revealed that three of CPAI factors (i.e., Social Potency, Dependability, and Accommodation) converged into the five factors of the NEO Five-Factor Inventory, whereas Interpersonal Relatedness factor was independent of the FFM dimensions [23–25]. The Interpersonal Relatedness factor contains the emic personality constructs that complement the etic constructs of CPAI-2. Previous studies using CPAI/CPAI-2 have shown its practicality as a culture-relevant personality measure in the Chinese context [26]. Here in this study, we attempt to perform GWAS to investigate the personality in the current Hakka people in Taiwan. We adopted CPAI-2 and aimed to better describe the personality of these Hakka people.

In Chinese, Hakka is a famous ethnic group for its history of large-scale migration. Migration and Hakka are two sides of the same thing. The origin of the name "Hakka" was given by the local Cantonese with the meaning of "guest". Leong [27] indicated the importance of migration and interaction with their neighbors to the formation of Hakka. Similarly, Hoerder [28] suggested internal Hakka migration in China up to the 1500s to be one of the "large scale and long-distance migrations", comparable to the Mongol expansion or Manchu penetration of China. On the other hand, Wang [29] suggested that Hakka's readiness to move is "characteristic of the Hakka migration pattern". In his viewpoint, Hakka's adventurous and pioneering spirit is one of the features that propel them to move compared to all the other dialect groups. He supplemented that the Hakka "were accustomed to working in remote areas throughout their history in China" [29] and hence, they simply migrated when the need arose. Leo [30] summarized these two studies and claimed that "migration" should be considered as a Hakka cultural marker. It should be the driving force that shaped and reshaped Hakka's identity.

Since the seventeenth century, and particularly during the eighteenth and nineteenth centuries, Hakka who came mainly from Guangdong agricultural communities emigrated to Taiwan, Malaya, and other regions of Southeast Asia, and as far as South Asia, Africa, Oceania, Europe, the Caribbean, and North and South America [31]. Among them, the Hakka who immigrated to Taiwan is mainly men. As the saying goes, only the father is from mainland China, and the mother is not, so marriage between the Chinese immigrants and the Taiwanese aboriginal population has been quite common since the invasion of Chinese, and the consequence is that a great number of Pepohoans (Plain's barbarians) have become assimilated in the manner of living to the immigrated Chinese peasants [32]. Strictly speaking, the current Hakka people in Taiwan are likely the descendants of a recent admixture between a group of immigrants with high emigration intention and a group of the aboriginal population generally without it.

Boneva and Frieze [33] proposed the term "migrant personality" that refers to psychological characteristics of individuals prone to emigration. They manifested that people who (1) are willing to do something challenging and unique (high achievement motive), (2) are willing to take the risk and endure dangers in reaching their goals (high power motive), and (3) show fewer concerns with emotional relationships or social network with families and friend (low affiliative motive), tend to have emigration intentions or behaviors. In case the personality, especially which related to the emigration intention, was indeed affected by some genetic factors, we would possibly find some of these phenotypic polymorphisms and the corresponding genetic polymorphisms in these Taiwanese Hakka people due to their special and recent immigration and admixture history. The associations between the phenotypic polymorphisms and genetic polymorphisms of nearby markers should have largely been maintained because the time elapsed since admixture is short and not enough for genetic recombination to break down their genetic linkages. Here we utilize these Hakka people to perform the GWAS. Since culture cannot change personality genetic make-up [34], we would like to know whether we could identify the genetic components of some personality constructs in CPAI-2 that were not previously observed using the traditional Big Five personality measures. To the best of our knowledge, this is the first study that investigates the biological basis of CPAI-2.

## Methods

### Participants

The current study is a sub-project of a large-scale cross-disciplinary research, which predominantly aimed to investigate the genetic origins of Taiwanese Hakka populations. Based on that specific purpose, the participants we recruited were restricted to individuals whose parents,

maternal grandparents, and paternal grandparents all speak the same certain Hakka dialect. The Hakka dialects include North Sixian, South Sixian, Hailu, Dabu, and Zhao'an, which are the most common Hakka dialects in Taiwan. We used this stringent criterion to ensure that the collected samples are representative enough without the disturbance from the frequent inter-population marriages in recent decades. Once one eligible participant was recruited in our study, his/her relatives (based on the family genealogy provided by the participant) would be excluded from the further recruitment. The saliva sample and personality questionnaire data were collected for the 288 recruited participants (159 males and 129 females). Our study protocols were approved by the Research Ethics Committee for Human Subject Protection of National Chiao Tung University/ IRB. The informed consent documents were also signed by all the participants. The ethical issue relating to usage institutional and government regulations is strictly followed within this study.

## Personality measurements

The Chinese Personality Assessment Inventory 2 (CPAI-2) [22] was used as the personality measure in this study. This Inventory consists of 22 normal personality scales and 3 validity scales. For this study, we used 19 personality scales including the Social Potency factor (i.e., Novelty, Diversity, Divergent Thinking, Leadership, Logical vs Affective Orientation, Extraversion vs Introversion, and Enterprise), Dependability factor (i.e., Responsibility, Emotionality, Interiority vs. Self-Acceptance, Optimism vs. Pessimism, Meticulousness, and Family Orientation), Accommodation factor (i.e., Veraciousness vs. Slickness), and Interpersonal Relatedness factor (i.e., Traditionalism vs. Modernity, Ren Qing, Social Sensitivity, Discipline, and Thrift vs. Extravagance). Five samples with incomplete questionnaire data were removed. CPAI-2 is a self-report measure and participants respond to the description of behavior in a true-false format. The sum scores for each scale were subsequently included in the GWAS analyses. We conducted a pilot study to examine its validity and reliability with 349 samples ($M_{age}$ = 37.73, $SD$ = 18.74). The items with no discriminant validity were deleted. The Cronbach's α values of the 19 personality scales ranged between 0.49 and 0.79 (the median Cronbach's α was 0.65) in the current study.

## Genotyping

A total of 547,644 SNPs were genotyped on the Illumina HumanCoreExome-24 v1.0 array from 288 samples in this study. Both the standard cluster and chromosome X cluster were conducted using GenomeStudio 2.0 [35] to genotype these SNPs. Four samples with insufficient genotyped SNPs (< 95%) were removed. A two-step quality control with six regulations was performed. The first step utilizing R programming (http://www.r-project.org/) removes (1) insertion / deletion polymorphisms (12,524 SNPs removed); (2) genotyped variants with the Chr. marker labeled as "0" (without the chromosome information; 803 SNPs removed), as "X", "Y", or "XY" (sex chromosome; 15,085 SNPs removed), or as "MT" (mitochondrial genome; 369 SNPs removed); (3) variants with missing rate > 5% (too many samples were not genotyped); and (4) positional duplicates (array SNPs with the same genomic position). A total of 11,364 SNPs were removed in (3) and (4). The second step utilizing PLINK [36] removes (5) variants with Hardy-Weinberg equilibrium (HWE) $p$ value $< 1 \times 10^{-5}$ (a deviation from the HWE that probably due to genotyping errors; 74 SNPs removed); and (6) variants with minor allele frequency (MAF) < 1% (without sufficient statistic power; 246,868 SNPs removed). A total of 260,557 SNPs were thus retained for further analyses.

## Genome-wide association testing

For each personality trait at each SNP locus, we used PLINK [36] to perform the linear regression analysis and calculate its probability value to clarify whether the personality scores from

the 279 participants are related to their genotypes. The standard $p$ value threshold $5 \times 10^{-8}$ was used. Quantitative trait association with 19 questionnaire score was conducted and the results were plotted into Manhattan and quantile-quantile plots with R. We also used LocusZoom [37] to draw their regional association plots. The power and sample size calculations were performed using genpwr [38] with linear and additive models.

## Results and discussion

### The identified genomic regions

The distribution patterns of the 19 personality scales for the 279 qualified participants were represented in S1 Fig, while the Manhattan plots for the 19 personalities were shown in Fig 1. The first significant result we found is a region flanked by *UTRN* and *EPM2A* located on chromosome 6 that is strongly associated with the personality, Enterprise (Fig 2A and Table 1). The protein encoded by *UTRN* is utrophin, which may play roles at neuromuscular junctions [39] and provide structural support for the neuronal membrane [40]. On the other hand, mutations of *EPM2A* may cause the malfunction of its encoded phosphatase, Laforin, leading to the accumulation of aberrant glycogen or polyglucosans and becoming neurotoxic [41]. This disorder is referred to as neurodegenerative Lafora disease. The *EPM2A* knockout mice were shown to have abnormal motor coordination, hind limb clasping, and even episodic memory deficits [41].

The second significant result is a region located in gene *NKAIN2* on chromosome 6 (Fig 2B and Table 1) that is strongly associated with Diversity. *NKAIN2* was previously reported relating to nervous system development [42]. One previous GWAS study showed that this gene is

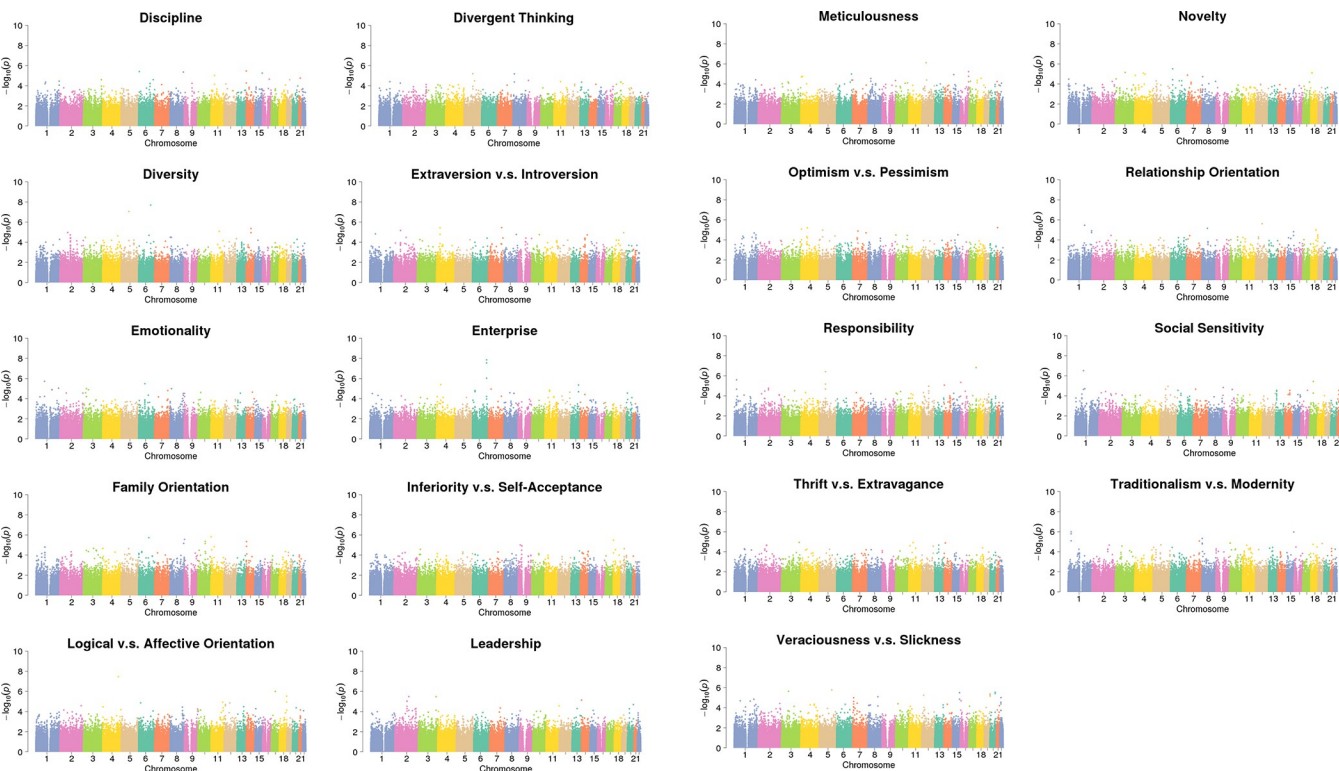

**Fig 1. The Manhattan plots of the probability values in linear regression analysis for the 19 personalities.** The horizontal axis in each plot represents the SNP location from chromosome 1 to chromosome 22, while the vertical axis indicates the significance level of that SNP (represented in $-\log_{10}$ format).

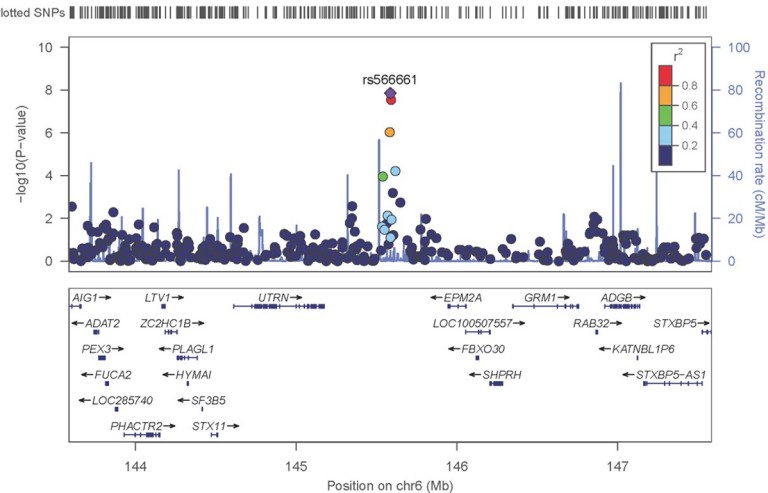

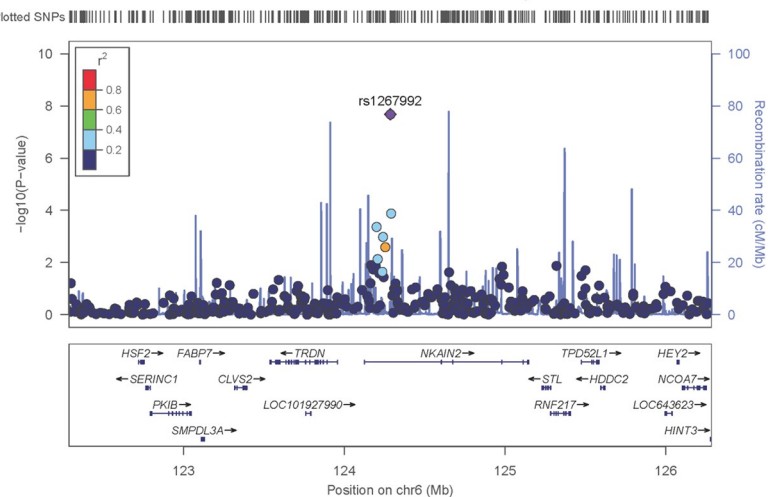

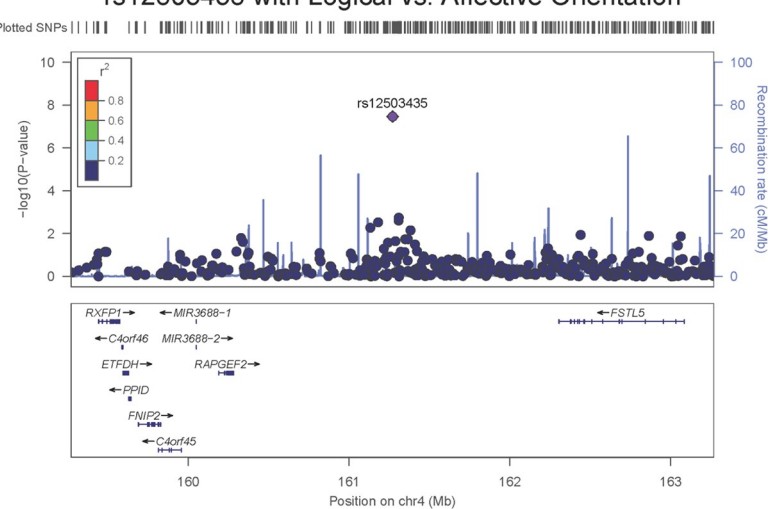

**Fig 2. Regional association plots around the three most significant SNPs.** (A) SNP rs566661 on chromosome 6 with Enterprise; (B) SNP rs1267992 on chromosome 6 with Diversity; (C) SNP rs12503435 on chromosome 4 with Logical vs. Affective Orientation. SNPs were plotted according to their probability values in the linear regression analysis (Fig 3 as the example) with the corresponding personalities. The purple rhombuses with SNP names are the most significant ones. The colors of the other circles represent their linkage disequilibrium ($r^2$) with the named top SNP. The genomic information denoted below was derived from human genome assembly GRCh37 (hg19) from Genome Reference Consortium.

associated with Neuroticism [43]; meanwhile, another GWAS study proposed that this gene is associated with Extraversion [8]. These results suggest that *NKAIN2* may have various effects on the development of personality.

The final significant result we found is a region flanked by *RAPGEF2* and *FSTL5* located on chromosome 4 that is strongly associated with Logical vs. Affective Orientation (Fig 2C and Table 1). The protein encoded by *RAPGEF2* is a neural-specific activator of Rap proteins [44, 45]. It is up-regulated in Alzheimer's disease patients' hippocampus and may play roles in Aβ oligomer-induced synaptic and cognitive degeneration [46]. *FSTL5* is expressed in cortical neurons [47], specifically in the hippocampus CA3 region and the cerebellum granular cell layer [48], and is important in synaptic transmission and plasticity [49]. It may also be related to obsessive-compulsive personality disorder [50].

## The interpretation of our GWAS findings

The classical QTL mapping methods usually conduct a hybridization process between individuals with distinct phenotypes for the feature of interest. As long as the frequencies of both the phenotype-causing variant and its neighboring genetic markers are significantly different between the two intermixing groups, linkage disequilibrium will be created even though the two groups are originally in linkage equilibrium. The joint segregation of phenotypic values and genetic markers distributed across the genome could thus be examined. In our study, the recent interethnic marriage between a group of immigrants (with high emigration intention) and a group of the aboriginal population (generally without the emigration intention) might coincidentally represent such a hybridization process, and thus create a situation similar to the artificial inter-cross populations and provide a superior opportunity to identify SNPs related to the emigration intentions and behaviors. The accomplishment of mapping the QTL in natural hybrid crosses is mainly determined by two factors: the sample size [51–53] and the duration of the time period [54]. Larger sample size may provide a larger statistical power; while a longer duration may increase the chance that the genetic linkage between the phenotypes and the genetic markers was broken by recombination events.

In recent years, most GWAS studies put their efforts into enlarging the sample size to increase the statistical power and the chance to successfully identify SNPs associated with traits of interest. For example, thousands to several hundred thousand individuals were usually

**Table 1. SNPs showing significant associations with one of the examined personality traits.**

| Chr | SNP | Position | Alleles | P value | | |
|---|---|---|---|---|---|---|
| | | | | Enterprise | Diversity | Logical vs. Affective Orientation |
| 6q24.3 | rs566661 | 145586542 | A/G | 1.420E-08* | 4.891E-03 | 1.359E-03 |
| 6q22.31 | rs1267992 | 124286626 | T/C | 3.622E-01 | 2.080E-08* | 2.929E-04 |
| 4q32.2 | rs12503435 | 161270471 | C/A | 4.677E-02 | 1.189E-02 | 3.440E-08* |

Chr, chromosome; SNP, single nucleotide polymorphism

* The p value thresholds for statistical significance: $5 \times 10^{-8}$.

included in the previously reviewed GWAS studies [7–10, 12–15], comparing to only 279 qualified participants in our study. Thus, some may argue that the sample size in our study is too limited. However, a small sample size could only generate Type 2 error and thus reduce the statistic power, but have nothing to do with Type 1 error [55]. In other words, a limited sample size would only reduce the chance to identify the character-related SNPs, but would not falsely recognize a SNP that is actually unrelated to the character. When performing a statistical test, the sample size issue has already been addressed. As long as the test is significant, the result should be reliable and not falsely generated; moreover, the impact of the identified variant must be strong to compensate for the small sample size effect and thus the identified variants should be relatively more important than the ones identified from a large sample set [55].

In our study, despite of the small sample size due to the stringent criterion of participant recruitments which unavoidably limits our statistical power, thanks to the special and recent immigration and admixture history of the Taiwanese Hakka people [32], we still found three genomic regions showing strong association with some personalities that might be related to the emigration intentions and behaviors (Table 1). Our results suggest that selecting an appropriate target population is also a crucial task. We utilized the $R^2$ values and genpwr [38] with linear and additive models to calculate the power for identifying the three most significant SNPs rs566661, rs1267992, and rs12503435 (Fig 3). The calculated values are 0.597, 0.571, and 0.049, respectively (S2 Fig). These values are not quite high and imply that there might be many other personality-related variants remaining unidentified due to our limited statistical power. The heritability contributed by these single variants could also be estimated from their allele frequencies and their effect sizes (defined as the regression coefficients) [56]. The estimated heritability values are 0.113, 0.105, and 0.050, which indeed supports the strong impacts of the identified variants on the analyzed Taiwanese Hakka populations.

It should be noted that the SNPs identified by GWAS are not necessarily themselves the phenotype-causing variants. In most cases, the association between the SNP and the feature is due to linkage disequilibrium. Moreover, the phenotype-causing variants may possibly be located in the regulatory regions instead of the coding regions, while the regulatory regions may still remain unknown. Therefore, we could only suggest that the phenotype-causing variant may occur in the neighboring areas but not a specific gene. Since multiple genes around the identified regions were reported to be related to the nervous system or expressed in the neural tissues, it is very likely that the personalities we examined are indeed controlled by some variants in these regions.

## The personalities identified in this study might correlate with the intention to emigrate

Among our identified results, Enterprise, Diversity, and Logical vs. Affective all belongs to the Social Potency factor in CPAI-2 [22]. Enterprise is defined as the degree to which individuals are ready to take risks; Diversity is defined as the degree to which individuals have diverse interests; and Logical vs. Affective Orientation is defined as the degree to which individuals are objective or subjective in their thinking and behavior [22]. While these three personality scales were jointly factor analyzed with the NEO-PI-R facets [23], Diversity and Logical vs. Affective Orientation loaded on Openness factor, and Enterprise loaded on both Neuroticism and Openness factors. In fact, Diversity and Logical vs. Affective Orientation are also the two personalities deviate mostly in the quantile-quantile plots (S3 Fig). Considering the special and recent immigration and admixture history of the Taiwanese Hakka people [32], we speculated that the successful identification of these three personality traits is probably related to the emigration intentions and behaviors of their ancestors.

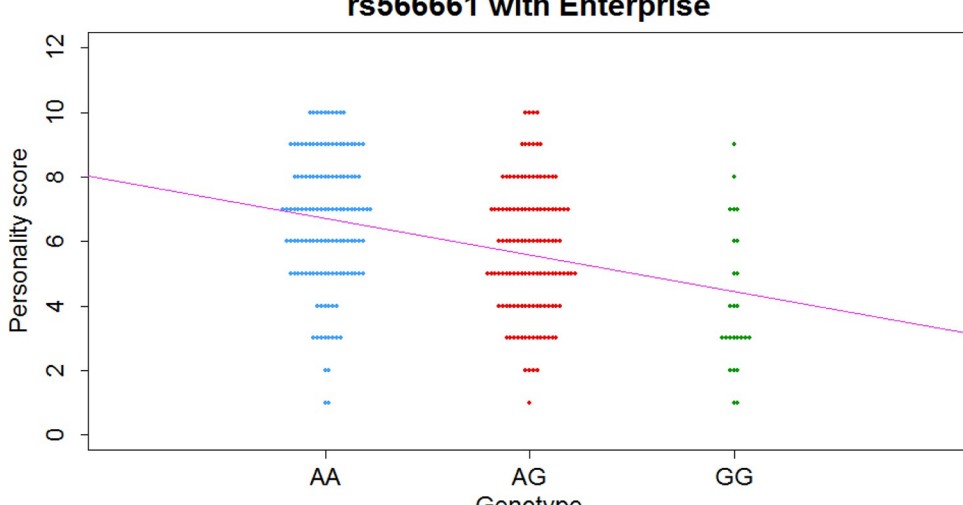

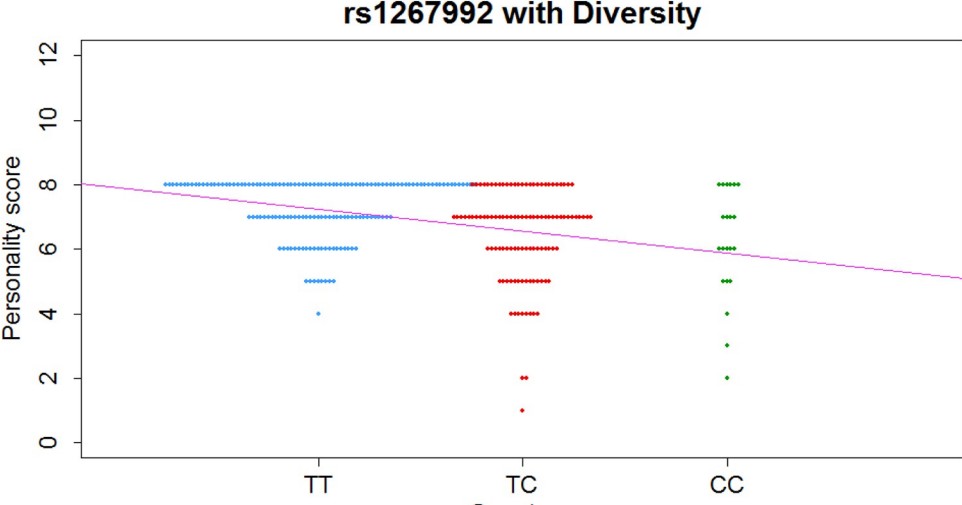

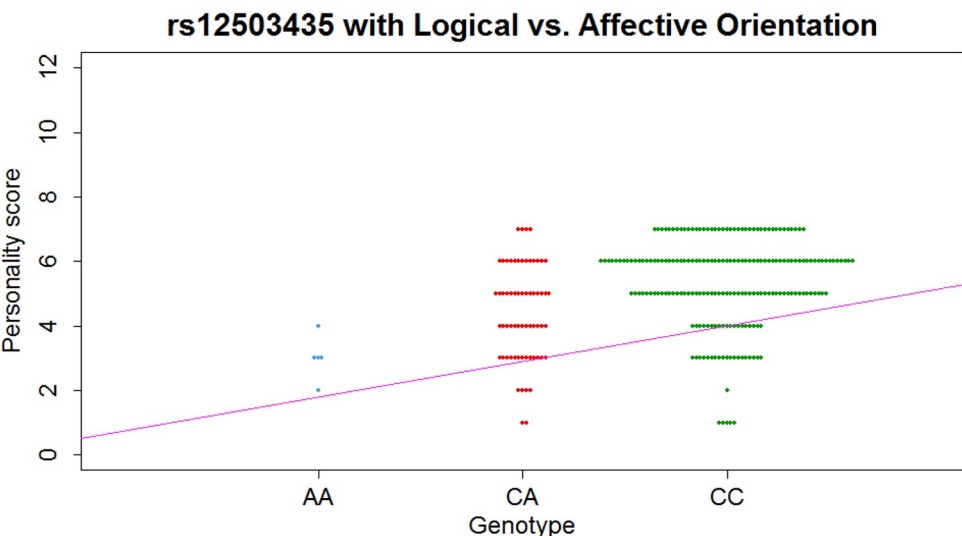

**Fig 3. The linear regressions between the personality score and genotype for the three most significant SNPs in Fig 2.** (A) SNP rs566661 on chromosome 6 with Enterprise. The Pearson correlation coefficient ($R$) is 0.331 and the regression coefficient ($\beta$, the slope) is -1.127; (B) SNP rs1267992 on chromosome 6 with Diversity ($R = 0.327$ and $\beta = -0.675$); (C) SNP rs12503435 on chromosome 4 with Logical vs. Affective Orientation ($R = 0.225$ and $\beta = 1.078$). The personality score and genotype of each participant were used to draw the swarm plots and obtain the regression lines.

Among the Big Five personality traits (Openness, Conscientiousness, Extraversion, Agreeableness, and Neuroticism), several previous studies have suggested that openness to experience is the most consistent one in predicting emigration intention [57–61]. McCrae and Costa [62] described open people as "are adventurous, bored by familiar sights, and stiffed by routine", and these characteristics are critical for self-selected migration. In contrast, conscientiousness trait was found to stably associate with lower desire to migrate [57, 58, 61, 63]. Conscientious people who tend to control impulses, comply with societal norms, and feel responsible for the home country have fewer intentions to leave [63]. Moreover, highly agreeable people were less likely to migrate because they have formed a stronger attachment with their homeland [64, 65]. In contrast, individuals who are extraverted and more emotionally stable displayed a stronger intention to migrate or work abroad [57–59, 61, 66]. Migration is fundamentally a bold move, and extraverted individuals are also active and have positive emotions [62]. They are more likely to socially interact with strangers and better adjust to the new land [59]. Other than the Big Five personality traits, Tabor et al. [61] further indicated that persistence and patience are also important characteristics of migrants because only the most persistent people would be able to successfully make it through the migration process.

Our result is generally consistent with these studies [57–61] that openness to experience is likely the most critical personality trait for emigration intentions and behaviors. We further suggest that the Social Potency factor should be the one most correlated with migrant personality in CPAI-2. However, it should be emphasized that, although the personalities mentioned above all belong to Openness and Social Potency factors, our results indicate that different facets in the same factor are controlled by different genetic variants (Table 1). For example, SNPs are significantly associated with one of the personality traits but not associated with the other two (Table 1). This implies that these genetic variants do not control the whole personality factor. Instead, they only independently control one of the facets in a personality factor defined in CPAI-2.

## The reliability of our GWAS findings

Independent statistical replications were frequently adopted to explore the reproducibility and reliability of GWAS studies [67]. Nevertheless, splitting the obtained genetic data into discovery and replication sets was unrecommended because it may decrease the statistic power and moreover, the replication set was argued to be not truly independent; therefore, most GWAS studies utilized data from mega-biobanks or meta-analysis of many other studies to perform the replication [67]. However, most GWAS studies and mega-biobanks are population-specific, and the examined traits are mostly restricted to common phenotypes, which may lead to the lack of available replication data for studies on non-European ancestry with seldom investigated traits and, in consequence, lead to the GWAS publication bias [67]. As expected, considering that we are the first study to investigate the genetic basis of CPAI-2 and considering the special immigration and admixture history of the Taiwanese Hakka people, there is no suitable external replication cohort available for implementation. In fact, this is inevitable for pioneer studies. In this circumstance, we alternatively pursued other biological lines of evidence to inspect the reliability of our identified results since these kinds of evidence should potentially be considered in the same vein as replications [67].

The first convincible evidence comes from the finding that all the three identified regions are either located in a neural gene or flanked by neural genes. It was recently reported that many adjacent neural genes are co-expressed and this co-expression is mainly controlled by distinct enhancers in the shared extended intergenic regions between these neighboring neural genes [68]. These extremely long intergenic regions of neural genes may contain numerous cell- and tissue-specific cis-regulatory elements which could provide an enormous amount of regulatory information to manipulate the complex and diverse gene expression patterns required in the mammalian nervous system [68]. If the personalities we examined indeed have genetic basis, their corresponding human behaviors should be controlled by the nervous system and thus these cis-regulatory elements should play important roles and have the potential to serve as the phenotype-causing variants.

Additional supporting evidence stems from the fact that the gene *NKAIN2* related to Diversity identified here was also reported to be associated with Neuroticism [43] and Extraversion [8] in previous GWAS studies. This could be considered as a different kind of replication since all the three studies independently confirm the important role of *NKAIN2* on the development of personality, although different personality traits and different genetic markers were recognized, which suggests that *NKAIN2* should have diverse functions in human behaviors. Based on the above descriptions, it is unlikely that our GWAS findings were derived just by chance or coincidence. The identified associations should thus provide valuable insights into the underlying genetic mechanisms of human behaviors and personalities.

## The mainlanders, islanders, and migrants

Camperio Ciani and colleagues [57, 58] compared the personality differences between island and mainland populations and found that mainlanders and immigrants from the mainland are significantly more open to experience than original islanders. They proposed that the difference was generated by a strong and non-random emigration flow from the islands–open people tend to emigrate and the sedentary islanders are thus comparatively less open. In our study, we did not directly compare the Chinese population and the Taiwanese aboriginal population (which is unavailable due to the frequent inter-population marriages in recent decades), but only analyzed their admixture descendants. Therefore, we could only speculate that the traits and alleles related to emigration intentions and behaviors were transported by the ancient Chinese immigrants, but could not draw a definite conclusion. In spite of that, we successfully identified three genomic regions significantly associated with potential migrant personalities. This result is consistent with the expectation inferred from Camperio Ciani and Capiluppi [58], i.e., the personality differences between immigrants and original islanders have genetic basis so that the admixture of them could generate a group of individuals with various personality tendencies, which is especially ideal for the GWAS studies.

## Prospect

It should be noted that the methods and strategies we utilized were not specific for the analysis of migrant personality. The key point is that the Taiwanese Hakka sample we used has such a special immigration and admixture history in the past 400 years [32]. Any genetic variants, especially genetic differences between the immigrants and aboriginal populations, have the potential to be identified through our GWAS analyses. Among them, features related to emigration intentions and behaviors should have the largest discrepancy and the highest chance to be recognized–no matter the difference was derived from the immigrants' high tendency, or from the sedentary islanders' low tendency. Once the sample size was enlarged and the statistical power was increased, we may potentially find more other SNPs associated with other

personalities not related to migration. Referring to the three SNPs identified in our study, the power to identify other SNPs with similar effect sizes could reach 0.9 as long as the sample sizes were enlarged to 400, 400, and 900, respectively (S2 Fig). It would be interesting to further investigate the genetic components of some CPAI-2 specific factors like Interpersonal Relatedness in Taiwanese populations.

## Supporting information

**S1 Fig. The distribution patterns of the 19 personality traits.**
(DOCX)

**S2 Fig. The plots of power by sample sizes for the three most significant SNPs shown in Fig 2.**
(DOCX)

**S3 Fig. The quantile-quantile plots for the 19 personalities.** The horizontal and vertical axes in each plot represent the theoretical and observed $p$-values in $-\log_{10}$ scale, respectively.
(DOCX)

## Acknowledgments

We are deeply grateful to all the study participants and numerous colleagues and friends who assisted with the recruitment process. We would like to thank Dr. Jenn-Kang Hwang for inspiring and initiating the project and Dr. Yan-Hwa Wu Lee for the continued supports. We also would like to appreciate the hardware supports from Dr. Tsun-Tsao Huang and the administrative supports from the Research Center for Humanities and Social Sciences, National Yang Ming Chiao Tung University.

## Author Contributions

**Conceptualization:** Yuh-Jyh Jong, Hsien-Da Huang.

**Data curation:** Ming-Hui Chen, Cheng-Yan Wang, Yih-Lan Liu.

**Formal analysis:** Pei-Ying Kao.

**Funding acquisition:** Wei-An Chang, Yuh-Jyh Jong.

**Investigation:** Pei-Ying Kao, Yeong-Shin Lin.

**Methodology:** Yih-Lan Liu.

**Project administration:** Wei-An Chang.

**Resources:** Mei-Lin Pan, Wei-Der Shu, Hong-Yan Chu, Cheng-Tsung Pan, Yih-Lan Liu.

**Supervision:** Wei-An Chang, Yih-Lan Liu, Yeong-Shin Lin.

**Validation:** Cheng-Yan Wang, Yih-Lan Liu, Yeong-Shin Lin.

**Visualization:** Pei-Ying Kao.

**Writing – original draft:** Pei-Ying Kao, Wei-An Chang, Mei-Lin Pan, Yih-Lan Liu, Yeong-Shin Lin.

**Writing – review & editing:** Pei-Ying Kao, Yeong-Shin Lin.

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
