## [Decision Letter · Decision Letter 0]

9 Nov 2022

PONE-D-22-24812A genome-wide association study (GWAS) of the personality constructs in CPAI-2 in Taiwanese Hakka populationsPLOS ONE Dear Dr. Lin

Thank you for submitting your manuscript to PLOS ONE. After careful consideration, we feel that it has merit but does not fully meet PLOS ONE’s publication criteria as it currently stands. Therefore, we invite you to submit a revised version of the manuscript that addresses the points raised during the review process.

We look forward to receiving your revised manuscript.

Kind regards,

Guanglin He

Academic Editor

PLOS ONE

Journal Requirements:

2. In the ethics statement in the Methods and online submission information, please ensure that you have specified (1) whether consent was informed and (2) what type you obtained (for instance, written or verbal, and if verbal, how it was documented and witnessed). If your study included minors, state whether you obtained consent from parents or guardians. If the need for consent was waived by the ethics committee, please include this information.

Reviewers' comments:

Reviewer's Responses to Questions

**Comments to the Author**

1. Is the manuscript technically sound, and do the data support the conclusions?

Reviewer #1: Yes

Reviewer #2: Yes

2. Has the statistical analysis been performed appropriately and rigorously? 

Reviewer #1: Yes

Reviewer #2: Yes

3. Have the authors made all data underlying the findings in their manuscript fully available?

Reviewer #1: Yes

Reviewer #2: Yes

4. Is the manuscript presented in an intelligible fashion and written in standard English?

Reviewer #1: Yes

Reviewer #2: Yes

5. Review Comments to the Author

Reviewer #1: In this manuscript, the authors used GWAS to identify the associations between genetic variants with the personality constructs in the CPAI-2 among Taiwanese Hakka population. With 279 participants collected, the authors reported three loci associated with Enterprise, Diversity, and logical vs affective orientation, respectively. These loci are located with nervous-related genes nearby. It’s a very meaningful and reasonable investigation, and the manuscript is written very well. However, I still have some concerns with the GWAS analysis:

Major concerns:

1) Although the authors discussed about the potential low power of the study due to the small sample size and emphasized the findings after the type 1 error rate control, I think it’s still good if the authors could provide some figures or tables of power calculated with respect to different sample sizes and effect sizes (Based on the range of effect sizes you observed from collected data).

2) I can understand that it’s difficult for authors to collect another independent study for replication, could you also provide an estimated power/sample size for replication studies? So that the readers can get a sense how many individuals are needed to replicate the findings. You can refer the following paper for details:

Jiang, W. and Yu, W., 2016. Power estimation and sample size determination for replication studies of genome-wide association studies. BMC genomics, 17(1), pp.19-32.

3) In the discussion section, there is one paragraph describing why the commonly used genome-wide significance level (5*10^-8) is not adopted in the manuscript. The authors stated that the extent of the (multiple testing) problem is actually positively correlated with the number of examined SNPs. I can’t agree with this statement in the GWAS setting. Taking account of the LD structures and number of haplotype blocks across the human genome, we can estimate an effective number of tests, that is, the equivalent number of independent SNPs along the genome. The genome-wide significance threshold was proposed not based on the number of SNPs tested in a given study but rather on the number of SNPs in the whole genome that might have been tested. You can refer the following paper for details. So I would suggest the authors changing the statement of this paragraph and reporting results based on genome-wide significance level. The authors can still report the findings passing the Bonferroni correction as strong-evidence findings. In the Bonferroni correction, we should also consider the number of phenotypes considered, so the adjusted threshold should be 0.05/(19*260557)=1e-8 .

Jannot, A.S., Ehret, G. and Perneger, T., 2015. P< 5× 10− 8 has emerged as a standard of statistical significance for genome-wide association studies. Journal of Clinical Epidemiology, 68(4), pp.460-465.

4) For the 19 phenotypes, could you provide an estimation of genetic correlations and phenotypic correlations? And, what’s the heritability estimated from the collected data?

5) If the phenotypes are genetically correlated, I would suggest the authors could consider conduct multi-trait analysis (e.g., MTAG) to improve power based on the summary statistics from each individual trait.

6) The SNP number after QC are almost half of the total collected SNP number. That’s unexpected. Could you also provide the filtered SNP number in each QC step? So that we can get a sense of the majority of SNPs were filtered due to which reason.

7) For association analysis, did you incorporate gender and PC (principal components) as covariates into the regression? If not, why not?

Reviewer #2: Distribution of Chinese personality Assessment Inventory 2 (CPA-2) of 288 recruited participants (159 males and 129 females) was not shown as histogram and the normality of the data we could not see for CPA-2. Authors should provide distribution pattern of CPA-2.

6. PLOS authors have the option to publish the peer review history of their article (what does this mean?). If published, this will include your full peer review and any attached files.

Reviewer #1: No

Reviewer #2: No

---

## [Author Response · Author response to Decision Letter 0]

24 Dec 2022

We responded to the reviewers' comments in the file "Response to reviewers.docx".

---

## [Editor Report · Decision Letter 1]

3 Feb 2023

Dear Dr. Lin,

We’re pleased to inform you that your manuscript has been judged scientifically suitable for publication and will be formally accepted for publication once it meets all outstanding technical requirements.

Kind regards,

Guanglin He

Academic Editor

PLOS ONE
---

## [Editor Report · Acceptance letter]

9 Feb 2023

PONE-D-22-24812R1 

A genome-wide association study (GWAS) of the personality constructs in CPAI-2 in Taiwanese Hakka populations 

Dear Dr. Lin:

I'm pleased to inform you that your manuscript has been deemed suitable for publication in PLOS ONE. Congratulations! Your manuscript is now with our production department. 

Kind regards, 

on behalf of

Dr. Guanglin He 

Academic Editor

PLOS ONE